# Measurement of the Intestinal pH in Mice under Various Conditions Reveals Alkalization Induced by Antibiotics

**DOI:** 10.3390/antibiotics10020180

**Published:** 2021-02-11

**Authors:** Kouki Shimizu, Issei Seiki, Yoshiyuki Goto, Takeshi Murata

**Affiliations:** 1Department of Chemistry, Graduate School of Science, Chiba University, Chiba 263-8522, Japan; 16SM3123@chiba-u.jp (K.S.); Issei.S@chiba-u.jp (I.S.); 2Medical Mycology Research Center, Division of Molecular Immunology, Chiba University, Chiba 260-8673, Japan; y-gotoh@chiba-u.jp; 3International Research and Development Center for Mucosal Vaccines, Division of Mucosal Symbiosis, Institute of Medical Science, The University of Tokyo, Tokyo 108-8639, Japan; 4Molecular Chirality Research Center, Chiba University, Chiba 263-8522, Japan

**Keywords:** antibiotics treatment, intestinal pH, mouse model, drug delivery, pre-clinical studies

## Abstract

The intestinal pH can greatly influence the stability and absorption of oral drugs. Therefore, knowledge of intestinal pH is necessary to understand the conditions for drug delivery. This has previously been measured in humans and rats. However, information on intestinal pH in mice is insufficient despite these animals being used often in preclinical testing. In this study, 72 female ICR mice housed in SPF (specific pathogen-free) conditions were separated into nine groups to determine the intestinal pH under conditions that might cause pH fluctuations, including high-protein diet, ageing, proton pump inhibitor (PPI) treatment, several antibiotic treatment regimens and germ-free mice. pH was measured in samples collected from the ileum, cecum and colon, and compared to control animals. An electrode, 3 mm in diameter, enabled accurate pH measurements with a small amount of gastrointestinal content. Consequently, the pH values in the cecum and colon were increased by high-protein diet, and the pH in the ileum was decreased by PPI. Drastic alkalization was induced by antibiotics, especially in the cecum and colon. The alkalized pH values in germ-free mice suggested that the reduction in the intestinal bacteria caused by antibiotics led to alkalization. Alkalization of the intestinal pH caused by antibiotic treatment was verified in mice. We need further investigations in clinical settings to check whether the same phenomena occur in patients.

## 1. Introduction

Intestinal pH is a crucial factor for drug delivery systems and pharmacokinetics [1,2]. Most drugs on the market are weakly basic: they dissolve in acidic pH but precipitate in alkaline conditions. Therefore, the intestinal pH is critical for drug solubility and affects the absorption and bioavailability, especially of oral drugs. In addition, enteric-coated drug delivery is increasingly being investigated [3,4,5,6]. Polymers dissolve depending on the pH and are used for modified or targeted drug release. Therefore, knowledge of the intestinal pH of humans and experimental animals is extremely important. The pH of the human gastrointestinal tract has previously been measured in healthy individuals and patients using a pH-sensitive wireless motility capsule [7,8,9].

Currently, rodents (mainly mice and rats) are often used in preclinical testing because of their small size and low cost. The gastrointestinal pH in rats has been reported under various conditions [10,11,12,13]. For example, the pH in their stomach has been measured after administering proton pump inhibitors (PPIs) [13]. However, we have incomplete information on the conditions in the gastrointestinal tract, especially in mice. The intestinal pH of mice has been reported only under normal conditions, and that of other mouse models is unavailable [10]. In this study, we measured the intestinal pH in mice on high-protein (HP) diets, aged mice, mice treated with PPIs, mice treated with antibiotics, and germ-free mice to identify factors influencing pH fluctuations.

## 2. Results and Discussion

### 2.1. Measurement of the Intestinal pH in Mice

In this study, we measured the intestinal pH in three parts of the gastrointestinal tract (ileum, cecum, and colon; Figure 1). Approximately 50–100 mg contents were used for pH measurements. We used all samples collected from the ileum and colon and a part of the sample from the cecum. The amount collected from the stomach and jejunum were too low to allow pH measurements, because the animals are fasted overnight before euthanasia. We added minimum water to the samples and performed the measurement on the smallest scale possible to maintain the original pH. The median (minimum−maximum) pH values of the ileum, cecum, and colon samples were 8.07 (7.93–8.20), 5.67 (5.57–5.96), and 5.87 (5.63–6.02), respectively (Table 1; Figure 2; control: ○ white circle). The pH values (median (minimum–maximum); ileum: 8.07 (7.93–8.20), colon: 5.87 (5.63–6.02)) are comparable to those (mean ± SD; ileum: 7.49 ± 0.46, colon: 6.63 ± 0.67) of humans reported in previous studies [7]. In the small intestine, the bicarbonates and bile contained in pancreatic juice neutralize the gastric acid, and the intestinal pH becomes alkaline. The pH of the upper part of the gut is greatly affected by digestive juices, and fewer samples were collected from these parts. The bottom part of the intestine is rich in bacteria; therefore, this pH information is important for colon-targeted drug delivery systems.

### 2.2. pH Measurement in Mice on High-Protein Diets

Intestinal bacteria digest proteins and generate harmful amines; therefore, a high-protein diet can affect the intestinal pH [14,15]. Previous studies have reported that rats on high-protein diets show variations in the proportion of the intestinal microbiome and increased inflammatory factors; however, the intestinal pH was not measured [16]. In this study, we measured the intestinal pH of mice that were fed a high-protein diet (containing 65% protein) for 1 week. The median (minimum−maximum) pH values of the ileum, cecum, and colon samples from these mice were 8.08 (7.85–8.25), 6.30 (6.11–6.67), and 6.43 (6.30–6.81), respectively (Table 1; Figure 2; HP: ◆ black diamond). The pH of the cecum and colon increased by approximately 0.5 compared to that in the control mice (*p* < 0.05), whereas the ileum pH remained almost the same. The increased intestinal pH was probably due to the amines generated from dietary proteins by the bacteria colonizing the cecum and colon.

### 2.3. pH Measurement in the Mice Treated with the Proton Pump Inhibitor

PPIs suppress the secretion of gastric acid and decrease the acid flowing into the intestinal tract. In addition, when the bactericidal effect of gastric acid weakens, bacteria that enter the gut increase and change the intestinal environment [17,18,19,20]. Previous research has demonstrated that the administration of a PPI significantly increases the pH of the stomach in humans and rats [13,21]. However, the influence of PPIs on the pH of the lower intestine is not known, and pH measurements in mice treated with PPIs have not been reported. In this study, we measured the intestinal pH of mice treated with 1 mM vonoprazan, a PPI, for 1 week. The median (minimum−maximum) pH values of the ileum, cecum, and colon samples were 7.60 (7.47–7.82), 5.54 (5.42–5.83), and 5.70 (5.30–5.93), respectively (Table 1; Figure 2; PPI: ▲ black triangle). Surprisingly, the pH decreased by approximately 0.5 in the ileum compared to the ileal pH in control mice (*p* < 0.05). The low level of gastric acid secretion might cause the increasing number of gut bacteria [17,20], resulting in a decrease in intestinal pH.

### 2.4. pH Measurement in Aged Mice

The composition of the intestinal microbiome changes with age in humans, and ageing is a factor responsible for the increasing number of pathogenic bacteria causing dysbiosis [22,23]. However, in a previous study, the intestinal pH of aged rats was not significantly different from that of young rats [12]. In this study, we measured the intestinal pH of elderly mice at 25–30 weeks of age. The median (minimum−maximum) pH values of their ileum, cecum, and colon samples were 8.04 (7.88–8.21), 5.83 (5.67–6.03), and 6.02 (5.81–6.26), respectively (Table 1; Figure 2; aged: × multiplication sign). These values were very similar to those of the control mice and consistent with the values previously reported in rats. This suggests that ageing is not an important factor affecting pH changes in mice.

### 2.5. pH Measurement in Mice Treated with Antibiotics

Antibiotics are administered to post-operative and hospitalized patients to control bacterial infections such as sepsis. In addition, antibiotics are often used in combination with other drugs [24,25,26,27]. As a result, the intestinal microbiome is significantly changed by antibiotics, and the intestinal pH can change due to the decrease in bacteria [28,29,30]. However, there is no precedent study in which intestinal pH has been measured when humans or experimental animals are treated with antibiotics.

In this study, we measured the intestinal pH in three groups of mice treated with different antibiotics: 0.5 g/L ampicillin (Amp), 0.5 g/L cefoperazone (Cpz), and 0.5 g/L vancomycin (Van) for 1 week. After antibiotic treatment, the gastrointestinal tract was darkened and expanded especially in the cecum in all groups compared to that in the control mice (Figure 1A, Figure 3A). Drastic pH increase was observed in the cecum and colon in all groups treated with antibiotics (*p* < 0.01), where the acidic pH became alkaline (Table 2; Figure 4; Amp/Cpz/Van: ◆ black diamond), although the solution of each of the three types of antibiotics was acidic pH.

To confirm the influence of antibiotics, we administered 0.5 g/L Amp to mice for 1 week and switched back to distilled water for the next week (Amp-DW). The darkened and expanded gastrointestinal tract returned to a normal color and size after switching back to distilled water (Figure 3B). In this group, the median (minimum−maximum) pH values of the ileum, cecum, and colon samples were 8.13 (8.01–8.33), 6.28 (6.03–6.62), and 6.38 (6.21–6.64), respectively (Table 2; Figure 4; Amp-DW: ◇ white diamond). The antibiotics exerted therapeutic effects, and the increasing pH almost returned to the normal value. Thus, the significant pH increase is probably the result of the antibiotic-induced decrease in intestinal bacteria.

### 2.6. pH Measurement in Germ-Free Mice

To verify the influence of commensal bacteria, we measured the intestinal pH of germ-free mice. The gut of the germ-free mice showed the same characteristics as that of the experimental mice treated with antibiotics (Figure 3C). The median (minimum−maximum) pH values of the ileum, cecum, and colon samples from germ-free mice were 8.44 (8.18–8.63), 7.66 (7.43–7.89), and 7.95 (7.72–8.14), respectively (Table 2; Figure 4; GF: ▲ black triangle). From these findings, we concluded that the decrease in or absence of intestinal bacteria causes a drastic pH increase in the lower intestine.

Moreover, due to the low concentration of oxygen, anaerobic bacteria typically increase toward the end of the intestine and occupy a large area. Commensal anaerobic bacteria degrade dietary fiber to produce short-chain fatty acids, including butyric acid, resulting in decreased intestinal pH [31,32,33]. The influence of gastric acid and digestive juice decreases along the ileum. Moreover, the increasing number of bacteria is the main factor affecting the pH change in the cecum and colon. They produce short-chain fatty acids that keep the intestine in acidic conditions to suppress the growth of pathogenic bacteria and promote intestinal peristaltic movement [33]. In our study, darkening and expansion of the intestinal tract occurred in mice treated with antibiotics and in GF mice. The absence of or decrease in intestinal bacteria causes indigestion, and peristaltic movement was attenuated.

## 3. Materials and Methods

### 3.1. Animals

In total, 64 female ICR mice (8–10 weeks of age) and eight retired mice (25–30 weeks of age) from Japan SLC, Inc., (Shizuoka, Japan) were used in this study. Mice were kept in a standard environment (23 °C and 12 h light–dark cycle) with free access to food and water; 1 week acclimatization period was set before the experiments.

### 3.2. Experimental Groups and Study Design

The mice were assigned to nine groups (eight mice per group) as follows. (1) Control: this group consisted of 8–10 week-old specific pathogen-free mice that were fed a basal diet (gamma ray-sterilized, containing 18.6% protein; PMI Nutrition International, Brentwood, MI, USA) and distilled water; (2) HP: this group was fed a high-protein diet (containing 64.0% protein; PMI) for 1 week; (3) PPI: this group was administered vonoprazan, a PPI (1 mM; Takeda, Tokyo, Japan), in drinking water for 1 week; (4) aged: this group included 25–30 week-old retired mice obtained from the supplier; (5) Amp: this group was administered ampicillin (0.5 g/L; Nacalai Tesque, Kyoto, Japan) in drinking water for 1 week; (6) Cpz: this group was administered cefoperazone (0.5 g/L, containing sulbactam; Teva Takeda Pharma Ltd., Nagoya, Japan) in drinking water for 1 week; (7) Van: this group was administered vancomycin (0.5 g/L; Pfizer, New York, NY, USA) in drinking water for 1 week. (8) Amp-DW: this group was administered ampicillin (0.5 g/L) for 1 week and switched back to distilled water for the next 1 week. (9) GF: this group included germ-free mice.

### 3.3. Dissection Procedure and pH Measurement

The mice were fasted overnight and sacrificed by cervical dislocation, and the intestinal tract was removed and divided into sections. The pH was measured in samples collected from the ileum, cecum, and colon. The contents of each section were removed, and 50 μL of purified water was added and mixed because most of the contents contained little water. The pH value was determined using a pre-calibrated portable pH meter LAQUA with a 9618S-10D Micro ToupH electrode (HORIBA, Ltd., Kyoto, Japan). This electrode, with a diameter of only 3 mm, can accurately measure the pH of a sample at a minimum volume of 50 μL. The measurement was performed immediately after dissection to minimize the influence of post-mortem changes. We measured the intestinal pH of eight mice per group while considering their individual differences.

### 3.4. Data Analysis

Statistical analyses of the data gathered from the animal experiments were conducted using Microsoft Excel (Microsoft Corporation, Redmond, WA, USA) and Prism ver.8 (GraphPad Software, San Diego, CA, USA). All values of the intestinal pH were plotted in the figures and expressed as the median (minimum value−maximum value) in the tables and main text. Differences between mouse groups were compared applying the Kruskal–Wallis test. Post hoc analysis was performed with Mann–Whitney U test with a Bonferroni correction applied. The pH values of the different gastrointestinal section of each group were compared applying the Friedman test. Post hoc analysis with Wilcoxon signed-rank tests was conducted with a Bonferroni correction applied. The results were considered statistically significant at *p* < 0.05.

## 4. Conclusions

Surprisingly, the pH of the cecum and colon increased drastically in the group treated with antibiotics. The marked effect on the intestinal microbiome after using antibiotics was reported in previous research [28,29,30]. The acidic intestinal pH became alkaline, and this result was consistent with the pH condition of GF mice. Since purified water was added to the contents, the real pH was higher for alkaline pH values. Clearly, the decreasing number of commensal bacteria has a critical influence on drug stability and solubility due to the increased intestinal pH and chronic intestinal obstruction. We analyzed the intestinal pH of female mice, and additional study is required to confirm this in male animals. Further clinical research is needed to verify whether the same phenomenon occurs in humans. Furthermore, we must optimize the conditions for the absorption and bioavailability of drugs concomitantly administered with antibiotics. Providing a suitable recovery period for patients is important even after discontinuing administration.

## Figures and Tables

**Figure 1 antibiotics-10-00180-f001:**
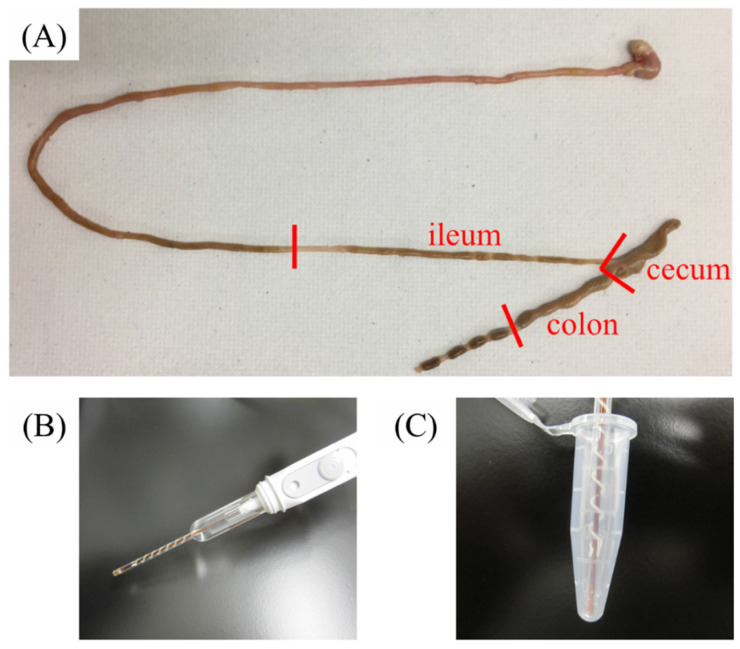
Sample collection and pH measurement. (**A**) Gastrointestinal tract of the control mouse. Samples were collected from three parts (ileum, cecum, and colon) of the gut of the mouse. (**B**) Micro ToupH electrode. (**C**) Electrode with a microtube that can measure a minimum volume of 50 μL.

**Figure 2 antibiotics-10-00180-f002:**
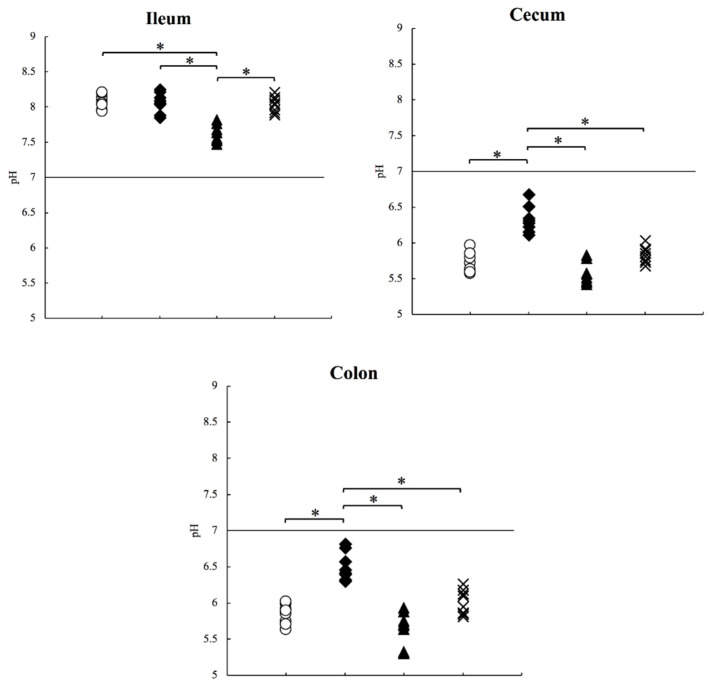
pH values measured in experimental mice under various conditions studied in the following groups. HP, the group with mice that were fed a high-protein diet (◆ black diamond); PPI, mice treated with the proton pump inhibitor (▲ black triangle); aged, elderly mice at 25–30 weeks of age (× multiplication sign), and control mice (○ white circle). * *p* < 0.05. In the ileum, the difference between HP vs. PPI > Control vs. PPI > Aged vs. PPI. In the cecum and colon, the difference between HP vs. PPI > HP vs. Control > HP vs. Aged.

**Figure 3 antibiotics-10-00180-f003:**
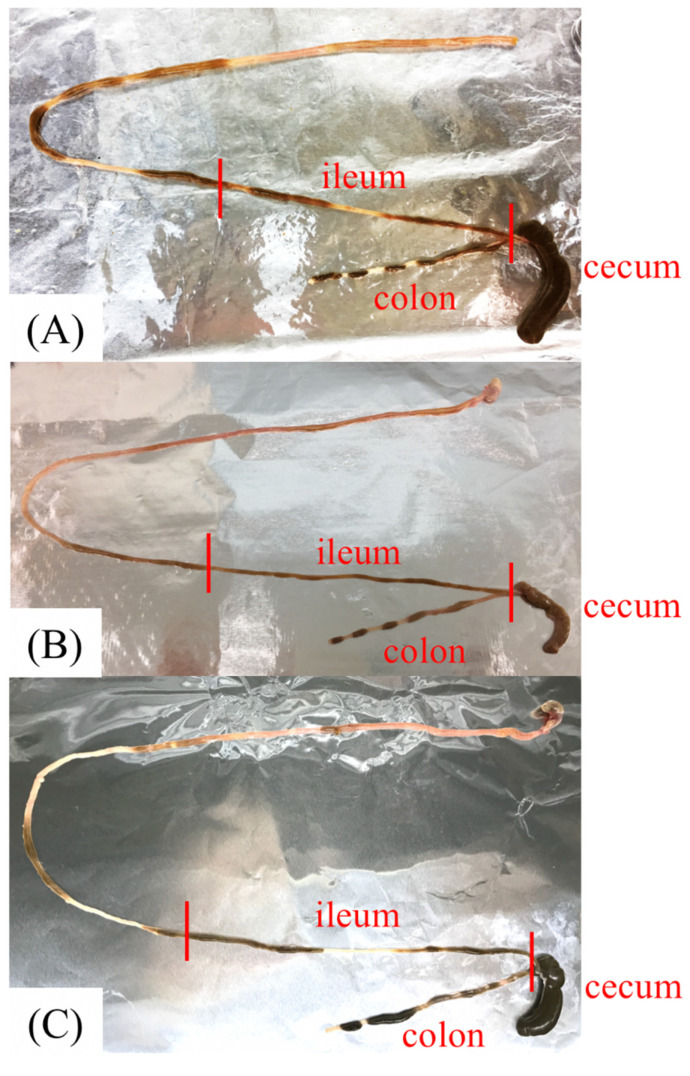
Gastrointestinal tract of the mice in various conditions. (**A**) Mouse treated for 1 week with ampicillin (Amp). (**B**) Mouse switched back to distilled water (DW) for 1 week after ampicillin treatment. (**C**) Germ-free mouse (GF).

**Figure 4 antibiotics-10-00180-f004:**
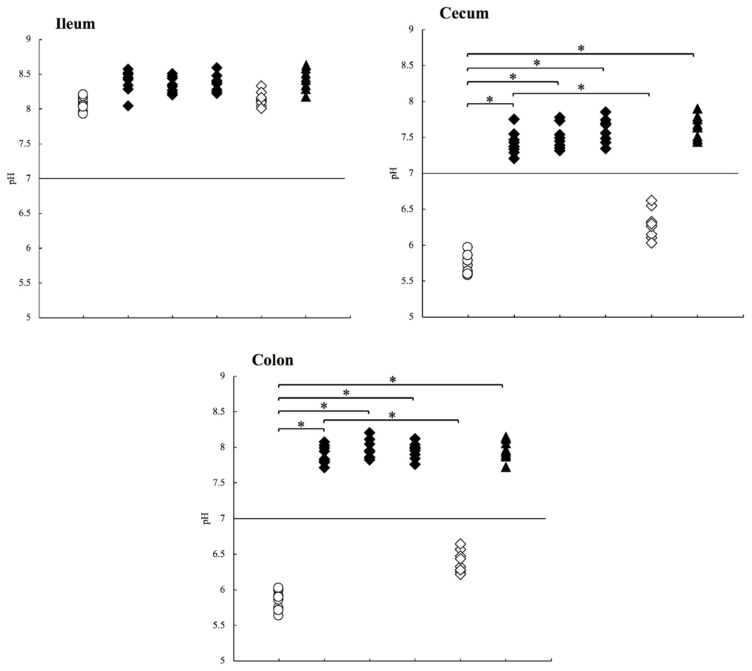
pH values measured in the mice treated with antibiotics studied in the following groups. The antibiotic groups included mice treated with antibiotics (Amp: ampicillin, Cpz: cefoperazone, Van: vancomycin, ◆ black diamond); Amp-DW, mice switched back to distilled water after ampicillin treatment (◇ white diamond); GF, germ-free mice (▲ black triangle) and control mice (○ white circle). * *p* < 0.01. In the cecum and colon, the differences between Control vs. Amp/Cpz/Van/GF > Amp vs. Amp-DW.

**Table 1 antibiotics-10-00180-t001:** pH values in different parts of mouse guts with different treatments, shown as the median (minimum value−maximum value). Since purified water (approximately pH 7) was added to contents, the real pH is higher for alkaline pH values and lower for acidic pH values. In all groups tested, pH values were significantly different (*p* < 0.05) between ileum vs. cecum and ileum vs. colon but not cecum vs. colon.

Mouse Group	Ileum	Cecum	Colon
**Control**	* 8.07 (7.93–8.20)	5.67 (5.57–5.96)	5.87 (5.63–6.02)
**HP**	* 8.08 (7.85–8.25)	6.30 (6.11–6.67)	6.43 (6.30–6.81)
**PPI**	* 7.60 (7.47–7.82)	5.54 (5.42–5.83)	5.70 (5.30–5.93)
**Aged**	* 8.04 (7.88–8.21)	5.83 (5.67–6.03)	6.02 (5.81–6.26)

HP, high protein; PPI, proton pump inhibitor. * *p* < 0.05 vs. cecum and colon.

**Table 2 antibiotics-10-00180-t002:** pH values in different parts of mouse guts with antibiotic treatment, shown as the median (minimum value−maximum value). Since purified water (approximately pH 7) was added to the contents, the real pH was higher for alkaline pH values and lower for acidic pH values. In all groups tested, pH values were significantly different (*p* < 0.05) between ileum vs. cecum, ileum vs. colon and cecum vs. colon (only Amp/Cpz/Van/GF groups).

Mice Group	Ileum	Cecum	Colon
**Control**	* 8.07 (7.93–8.20)	5.67 (5.57–5.96)	5.87 (5.63–6.02)
**Amp**	* 8.44 (8.05–8.57)	** 7.40 (7.20–7.75)	7.89 (7.71–8.07)
**Cpz**	* 8.34 (8.20–8.51)	** 7.47 (7.31–7.77)	7.94 (7.82–8.20)
**Van**	* 8.38 (8.22–8.59)	** 7.61 (7.34–7.85)	7.97 (7.76–8.12)
**Amp-DW**	* 8.13 (8.01–8.33)	6.28 (6.03–6.62)	6.38 (6.21–6.64)
**GF**	* 8.44 (8.18–8.63)	** 7.66 (7.43–7.89)	7.95 (7.72–8.14)

Amp, ampicillin; Cpz, cefoperazone; Van, vancomycin; Amp-DW, mice treated with Amp for 1 week and switched back to distilled water for 1 week; GF, germ-free. * *p* < 0.05 vs. cecum and colon. ** *p* < 0.05 vs. colon.

## Data Availability

Not applicable.

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
