# Peer review of "Measurement of the Intestinal pH in Mice under Various Conditions Reveals Alkalization Induced by Antibiotics"

_antibiotics, 2021, doi:10.3390/antibiotics10020180_

Round 1
Reviewer 1 Report
I have already revised the manuscript of Shimizu et al. in the previous submission, and although I appreciated the efforts of the authors in doubling the statistical population some clarifications are necessary and important:
- Line 202-203: "(4) aged: this group included 25–30 week-old retired mice obtained from the supplier" however in the previous submission it was indicated "(4) Aged: this 179 group was housed in a standard environment with free access to basal diet and distilled 180 water until 30 weeks of age." So, these are different conditions, which is the real one of the first presented group here combined with the new second group. Two groups subjected to different conditions cannot be combined.
- How long did acclimatization last before each treatment? It should be reported.
- If the authors change the statistical evaluation from t-student to one-way analysis of variance, also the statistical indications in the figures should be changed. I mean, the hierarchy of significance among the means should be reported, e.g. with letters, from the most significant to the least significant difference. Reported as now with connectors is a comparison one group against one group, as it occurs in t-student test.
- Lines 214-215 and lines 53-55: "We could not collect sufficient samples to measure the pH in several sections of the stomach and upper small intestine." "Low amounts of contents (< 10 mg) were collected from the stomach and jejunum, because mice are fasted overnight before euthanasia. The pH values of the stomach measured as follows were not accurate compared to other pH values, as a reference." from this is not clear if: (1) the pH measurement was made on all animals, but with small amounts (<10 mg), (2) the pH measurement was done only in some groups (showed) and with small amounts (<10 mg). In the case (1) the gastric pH values of the other groups should be shown together with the minimum amount of samples considered; in the case (2) the n of animals for each measurement should be shown together with the minimum amount of samples considered.
Author Response
Response to Reviewer 1 Comments
*Line ?? is refer to revised PDF file.
Comment 1-1:Line 202-203: "(4) aged: this group included 25–30 week-old retired mice obtained from the supplier" however in the previous submission it was indicated "(4) Aged: this 179 group was housed in a standard environment with free access to basal diet and distilled 180 water until 30 weeks of age." So, these are different conditions, which is the real one of the first presented group here combined with the new second group. Two groups subjected to different conditions cannot be combined.
A:Thank you for your comment. We performed pH measurement in “eight” retired mice (25–30 weeks old) from the supplier in additional experimentsand replaced the results.
Comment 1-2:How long did acclimatization last before each treatment? It should be reported.
A:Thank you for your suggestion. We added the sentence: 1 week acclimatization period was set before the experiments (Line 191).
Comment 1-3:If the authors change the statistical evaluation from t-student to one-way analysis of variance, also the statistical indications in the figures should be changed. I mean, the hierarchy of significance among the means should be reported, e.g. with letters, from the most significant to the least significant difference. Reported as now with connectors is a comparison one group against one group, as it occurs in t-student test.
A:Thank you for your advice. We compared each group and all significant differences were shown in the Figure 2 & 4 and their legends.
Comment 1-4:Lines 214-215 and lines 53-55: "We could not collect sufficient samples to measure the pH in several sections of the stomach and upper small intestine." "Low amounts of contents (< 10 mg) were collected from the stomach and jejunum, because mice are fasted overnight before euthanasia. The pH values of the stomach measured as follows were not accurate compared to other pH values, as a reference." from this is not clear if: (1) the pH measurement was made on all animals, but with small amounts (<10 mg), (2) the pH measurement was done only in some groups (showed) and with small amounts (<10 mg). In the case (1) the gastric pH values of the other groups should be shown together with the minimum amount of samples considered; in the case (2) the n of animals for each measurement should be shown together with the minimum amount of samples considered.
A:Thank you for your suggestion.Actually, (2) the pH measurement was done only in some groups. The number of mice used in the measurement is same as other experiments. According to Reviewer 2’s Comment 2-6 and 2-13, we removed the results about pH measurementin the stomach. We left the statement in the text to showthat there were not enough contents to perform accurate pH measurements in the stomach and jejunum.
Reviewer 2 Report
The authors obviously have performed more experiments and used the correct statistical tests. The quality of the manuscript has much improved. However, I still have some comments and questions.
- Line 17: SPF mice, the strain of the mice should be shown, and the mice “were housed in SPF conditions”
- The nine groups should be named in the Abstract (line 17).
- In the Abstract, the authors should show their main findings: pH changes by high protein diet in the cecum and colon; pH changes by PPI in ileum, changes by treatment with antibiotics and findings in germ free mice.
- Abstract, line 19: “mainly” should be removed
- Introduction: The statement in line 41 needs a reference.
- Results 2.1: WE do not know how many measurements could be performed in the stomach and jejunum. Therefore, I suggest to remove these results completely from the study and just leave the statement in the text that there not enough contents to perform measurements in the stomach and jejunum.
- Table 1: Stomach and jejunum samples should be removed and if the authors show these date, then the median should be shown and not just a range.
- Table 2: The pH values of the different gastrointestinal section of each group (control, HP, PPI and aged) should be statistically compared (Friedman test as these are paired data)
- I think the > and < signs should be removed from the tables and the authors should show what the really measured. Still, the authors are right with there statement regarding the addition of water and therefore it should be left in text.
- Results, part 2.2: the statistically significant differences to the control group should be described in this section (line 84).
- For Figure 2, the authors should show all statistically significant differences found. For example, if controls and PPI were significantly different, then PPI and HP should have been also (just by looking at the figure).
- Line 107: show range as well
- Again, please state statistical significant differences in section 2.3 (line 110) and remove the gastric pH measurements.
- Line 110: “ileal” instead of “intestinal”
- Line 111: needs a reference.
- Line 128: “precedent study”
- Line 131: sentence is misleading, not each group was treated with all three antibiotics.
- Values in lines 134 to 138: the values do not have to be enumerated here since they are shown in Table 3, this is redundant.^
- Statement in Line 139: not true for ileum, when I look at the Figure
- Line 141: state the statistical differences
- Table 3: The pH values of the different gastrointestinal section of each group should be statistically compared (Friedman test as these are paired data)
- Lines 175 to 182: references are needed.
- Line 184 to 187: no data are shown, these lines should therefore be removed
- Line 241: phenomenon
- Materials and Methods, line 194: What do the authors mean with “since March 20th 2019”. In the next sentence they authors state that approval started from April 2019.
- Materials and Methods, line 197: how many animals per group? Pleas correct your statement “four” here and
- Materials and Methods: If the authors used aged mice (25-30 weeks) for the experiments the statement about age in line 190 is not correct.
- Material and Methods, line 214: “We could not collect sufficient samples….”
Author Response
Response to Reviewer 2 Comments
*Line ?? is refer to revised PDF file.
Comment 2-1:Line 17: SPF mice, the strain of the mice should be shown, and the mice “were housed in SPF conditions”
A:Thank you for your comment. We corrected the sentence as you suggested.
Comment 2-2:The nine groups should be named in the Abstract (line 17).
A:According to your suggestion, we revised in the Abstract (Line 18-19).
Comment 2-3:In the Abstract, the authors should show their main findings: pH changes by high protein diet in the cecum and colon; pH changes by PPI in ileum, changes by treatment with antibiotics and findings in germ free mice.
A:We agree. We added them in the Abstract (Line 21-23).
Comment 2-4:Abstract, line 19: “mainly” should be removed
A:The word was removed as you suggested.
Comment 2-5:Introduction: The statement in line 41 needs a reference.
A:Thank you for your advice. A suitable reference (13) was added (Line 43).
Comment 2-6:Results 2.1: WE do not know how many measurements could be performed in the stomach and jejunum. Therefore, I suggest to remove these results completely from the study and just leave the statement in the text that there not enough contents to perform measurements in the stomach and jejunum.
A:Thank you for this suggestion. We removed the results about the pH of stomach and left the statement in the text.
Comment 2-7:Table 1: Stomach and jejunum samples should be removed and if the authors show these data, then the median should be shown and not just a range.
A:We agree to remove Table 1.
Comment 2-8:Table 2: The pH values of the different gastrointestinal section of each group (control, HP, PPI and aged) should be statistically compared (Friedman test as these are paired data)
A:Thank you for your advice. The differences between Ileum vs Cecum, Ileum vs Colon, and Cecum vs Colon were compared using Friedman test followed by Mann-Whitney U test with Bonferroni’s correction (Line 224). All significant differences were shown in the legend of Table 1.
Comment 2-9:I think the > and < signs should be removed from the tables and the authors should show what the really measured. Still, the authors are right with there statement regarding the addition of water and therefore it should be left in text.
A:We removed the signs of inequality and left the statement in the legend.
Comment 2-10:Results, part 2.2: the statistically significant differences to the control group should be described in this section (line 84).
A:The statistical difference between Control vs HP was indicated (Line 82).
Comment 2-11:For Figure 2, the authors should show all statistically significant differences found. For example, if controls and PPI were significantly different, then PPI and HP should have been also (just by looking at the figure).
A:Thank you for your comment. We have shown all statistically significant differences found, just by looking at the Figure 2.
Comment 2-12:Line 107: show range as well
A:This pH value of the stomach was removed according to your Comment 2-6 and 2-13.
Comment 2-13:Again, please state statistical significant differences in section 2.3 (line 110) and remove the gastric pH measurements.
A:The statistical difference between Control vs PPI was indicated, and the pH values of stomach were removed from the section.
Comment 2-14:Line 110: “ileal” instead of “intestinal”
A:We corrected the word as you suggested.
Comment 2-15:Line 111: needs a reference.
A:Thank you for your advice. Suitable references (17, 20) were added.
Comment 2-16:Line 128: “precedent study”
A:We corrected the phrase as you suggested.
Comment 2-17:Line 131: sentence is misleading, not each group was treated with all three antibiotics.
A:We rephrased the sentence according to your comment (Line 130-132).
Comment 2-18:Values in lines 134 to 138: the values do not have to be enumerated here since they are shown in Table 3, this is redundant.
A:Thank you for your comment. We removed the values as your suggestion.
Comment 2-19:Statement in Line 139: not true for ileum, when I look at the Figure
A:Thank you for your advice. We removed this sentence.
Comment 2-20:Line 141: state the statistical differences
A:The statistical difference between Control vs antibiotics treatment was indicated (Line 135).
Comment 2-21:Table 3: The pH values of the different gastrointestinal section of each group should be statistically compared (Friedman test as these are paired data)
A:Thank you for your advice. The differences between Ileum vs Cecum, Ileum vs Colon, and Cecum vs Colon were compared using Friedman test followed by Mann-Whitney U test with Bonferroni’s correction (Line 224). All significant differences were shown in the legend of Table 2.
Comment 2-22:Lines 175 to 182: references are needed.
A:Thank you for your advice. The suitable references (31-33) were added in this section (Line 178, 182).
Comment 2-23:Line 184 to 187: no data are shown, these lines should therefore be removed
A:We removed the sentence as you suggested.
Comment 2-24:Line 241: phenomenon
A:We corrected the word as you suggested.
Comment 2-25:Materials and Methods, line 194: What do the authors mean with “since March 20th 2019”. In the next sentence they authors state that approval started from April 2019.
A:Thank you for your comment. We added some words for clarity. The Animal Research Ethics Committee approved the experiments in the end of March, and we were permitted to start the experiments from April. According to Reviewer 3’s Comment 3-3, we moved this section to Institutional Review Board Statement (Line 243).
Comment 2-26:Materials and Methods, line 197: how many animals per group? Pleas correct your statement “four” here and
A:We corrected “four” to “eight” (Line 193).
Comment 2-27:Materials and Methods: If the authors used aged mice (25-30 weeks) for the experiments the statement about age in line 190 is not correct.
A:Thank you for your comment. We rephrased the sentence in this section (Line 188).
Comment 2-28:Material and Methods, line 214: “We could not collect sufficient samples….”
A:We removed the sentence as you suggested.
Reviewer 3 Report
In the manuscript, Shimizu et al. describe the impact of protein pump inhibitor, aging, and antibiotics on intestinal pH in mice. This well written and illustrated article will be of broad interest to readers of antibiotics journal. However, some items need to be addressed before publication:
- My primary concern is the lack of one control experiment. It would be nice for authors to test the intestinal pH in Germ-Free Mice with antibiotic treatments. If no pH change occurs, it can exclude the possibility that other factors trigger the alkalization.
- Line 61: The authors claim their pH measurements are similar to those of humans. However, the ileum's pH values (7.49±0.49) reported in reference 7 are lower than the pH values reported in this manuscript; the colon's pH values (6.63±0.67) are higher than pH values reported here. The authors are suggested to rephrase the sentence.
- The institutional review board statement is missing between author contributions and funding sections. Authors are suggested to move their statement from line 192 to 195 to back-matter.
Author Response
Response to Reviewer 3 Comments
*Line ?? is refer to revised PDF file.
Comment 3-1:My primary concern is the lack of one control experiment. It would be nice for authors to test the intestinal pH in Germ-Free Mice with antibiotic treatments. If no pH change occurs, it can exclude the possibility that other factors trigger the alkalization.
A: Thank you for your suggestion. We understand your concern. However, germ-free mice are much more expensive than normal SPF mice. It is difficult for us to perform further experiments using GF mice due to its high cost. We added some supplementary explanation in the revised text instead (Line 134, 144, 172). We would appreciate your kind understanding on this matter.
Comment 3-2:Line 61: The authors claim their pH measurements are similar to those of humans. However, the ileum's pH values (7.49±0.49) reported in reference 7 are lower than the pH values reported in this manuscript; the colon's pH values (6.63±0.67) are higher than pH values reported here. The authors are suggested to rephrase the sentence.
A:Thank you for your advice. The sentence was rephrased (Line 59).
Comment 3-3:The institutional review board statement is missing between author contributions and funding sections. Authors are suggested to move their statement from line 192 to 195 to back-matter.
A:Thank you for your comment. The statementwas moved to the Institutional Review Board Statement between author contributions and funding sections (Line 243).
Round 2
Reviewer 1 Report
The authors addressed my comments except for statistic, and for the use of water as a limitation of the study to known the real pH values in the conclusion. I suggest to perform these changes before publication.
Author Response
Response to Reviewer 1 Comments
*Line ?? is refer to revised PDF file.
Comment 1-1:The authors addressed my comments except for statistic, and for the use of water as a limitation of the study to known the real pH values in the conclusion. I suggest to perform these changes before publication.
A:Thank you for your comment. For statistic, we compared each group and showed with inequality sign in the legend of Figure 2 & 4. For the use of water as a limitation of the study, we added the sentence in the conclusion (Line 233) and the legend of Table 1 & 2 (Line 89, 161).
Reviewer 2 Report
The authors have implemented the changes according to the suggestions. The quality has much improved. I have a few remaining comments which should be addressed:
- Abstract, line 18: “…proton pump inhibitor (PPI) treatment, several antibiotic treatment regimens and germ-free mice.”
- Abstract, line 20: “…cecum and colon and compared to control animals.”
- Results and Discussion, line 54: Rephrase the sentence “Low amounts of contents (< 10 mg) were collected from the stomach and jejunum, because mice are fasted overnight before euthanasia” to something like “The amount collected from the stomach and jejunum were too low to allow pH measurements because the animals were fasted overnight before euthanasia.”
- Results and Discussion, line 59: Were this mean or median values? Please add the appropriate term: “The mean/median pH values of the ileum….were 8.07….”. And what are the values in parentheses? Range? SD? Please also add this information. Same for values in lines 61 and 62 and in lines 84 and 85 and lines 113-114, line 125, lines 154ff, line 184 (I assume the authors show median and range, but please add this information in the text).
- Table 1: Statement, line 93ff: Change the statement to: “In all groups tested pH values were significantly different (p<0.05) between ileum vs. cecum and ileum vs. colon but not cecum vs. colon.” And please add a sign (* for example) following the values of the ileum (in all 4 groups) and add a “*…p<0.05 vs. cecum and colon” to the Table Legend.
- Concerning the statistics: The authors performed a Friedmann Test for testing for global significances for the gastrointestinal sections of the different groups. However, in this case post hoc analysis should be performed with Wilcoxon Tests with a Bonferroni correction applied. And please state in the Statistics section: “Differences between the groups were compared applying the Kruskal-Wallis test. Post hoc analysis was performed with Mann Whitney-U tests with a Bonferroni correction applied. The pH values of the different gastrointestinal section of each group were compared applying the Friedman test. Post hoc analysis with Wilcoxon signed-rank tests was conducted with a Bonferroni correction applied.”
- Figure Legend, Figure 2: The “Each group was compared and all significant differences were shown:” can be removed. Also the “PPI vs Control/HP/Aged in the ileum; HP vs Con-102 trol/PPI/Aged in the cecum and colon”. It is sufficient to just say “*…p<0.05”
- Line 147: Remove “In particular”
- Figure Legend, Figure 4: The “Each group was compared and all significant differences were shown:” can be removed. Also the last sentence of the Figure Legend can be removed. It is sufficient to just say “*…p<0.01”
- Table 2: Statement, line 174ff: Please also apply the suggested changes for Table 1 (see above) to this Table.
Author Response
Response to Reviewer 2 Comments
*Line ?? is refer to revised PDF file.
Comment 2-1:Abstract, line 18: “…proton pump inhibitor (PPI) treatment, several antibiotic treatment regimens and germ-free mice.”
A:Thank you for your comment. We corrected the sentence as you suggested.
Comment 2-2:Abstract, line 20: “…cecum and colon and compared to control animals.”
A:We corrected the sentence as your suggestion.
Comment 2-3:Results and Discussion, line 54: Rephrase the sentence “Low amounts of contents (< 10 mg) were collected from the stomach and jejunum, because mice are fasted overnight before euthanasia” to something like “The amount collected from the stomach and jejunum were too low to allow pH measurements because the animals were fasted overnight before euthanasia.”
A:According your comment, we corrected the sentence (Line 54).
Comment 2-4:Results and Discussion, line 59: Were this mean or median values? Please add the appropriate term: “The mean/median pH values of the ileum….were 8.07….”. And what are the values in parentheses? Range? SD? Please also add this information. Same for values in lines 61 and 62 and in lines 84 and 85 and lines 113-114, line 125, lines 154ff, line 184 (I assume the authors show median and range, but please add this information in the text).
A:Thank you for your advice. We added “median (minimum − maximum)” before each part of pH value, and “mean ± SD” before pH value from the reference (Line 57, 60, 61, 80, 108, 119, 143, 170).
Comment 2-5:Table 1: Statement, line 93ff: Change the statement to: “In all groups tested pH values were significantly different (p<0.05) between ileum vs. cecum and ileum vs. colon but not cecum vs. colon.” And please add a sign (* for example) following the values of the ileum (in all 4 groups) and add a “*…p<0.05 vs. cecum and colon” to the Table Legend.
A:Thank you for your suggestion. We rephrased the legend and corrected Table 1.
Comment 2-6:Concerning the statistics: The authors performed a Friedmann Test for testing for global significances for the gastrointestinal sections of the different groups. However, in this case post hoc analysis should be performed with Wilcoxon Tests with a Bonferroni correction applied. And please state in the Statistics section: “Differences between the groups were compared applying the Kruskal-Wallis test. Post hoc analysis was performed with Mann Whitney-U tests with a Bonferroni correction applied. The pH values of the different gastrointestinal section of each group were compared applying the Friedman test. Post hoc analysis with Wilcoxon signed-rank tests was conducted with a Bonferroni correction applied.”
A:Thank you for your advice. We performed the statistics with Wilcoxon tests with a Bonferroni correction for post hoc analysis, and corrected the sentences in the Statistics section (Line 223).
Comment 2-7:Figure Legend, Figure 2: The “Each group was compared and all significant differences were shown:” can be removed. Also the “PPI vs Control/HP/Aged in the ileum; HP vs Con-102 trol/PPI/Aged in the cecum and colon”. It is sufficient to just say “*…p<0.05”
A:Thank you for your comment. We removed these sentences from the legend of Figure.
Comment 2-8:Line 147: Remove “In particular”
A:We removed the word (Line 136).
Comment 2-9:Figure Legend, Figure 4: The “Each group was compared and all significant differences were shown:” can be removed. Also the last sentence of the Figure Legend can be removed. It is sufficient to just say “*…p<0.01”
A:Thank you for your comment. We removed these sentences from the legend of Figure.
Comment 2-10:Table 2: Statement, line 174ff: Please also apply the suggested changes for Table 1 (see above) to this Table.
A:Thank you for your suggestion. We rephrased the legend and corrected Table 2.
Reviewer 3 Report
The authors have addressed my concerns. The revised manuscript looks good.
Author Response
Response to Reviewer 3 Comments
Comment 3-1:The authors have addressed my concerns. The revised manuscript looks good.
A: Thank you very much for your comment.
Round 3
Reviewer 2 Report
All concerns have been answered. The are some minor issue. But these can be corrected during editorial checking.
This manuscript is a resubmission of an earlier submission. The following is a list of the peer review reports and author responses from that submission.
Round 1
Reviewer 1 Report
The manuscript of Shimizu et al. analyzed the pH changes in the GI of mice subjected to various treatments. This is a well described study, I only have some suggestions in order to improve the manuscript.
- the authors analyzed only pH of female mice, and this should be indicated in the conclusions as a limitation.
- the addition of water, although made for all samples, did not allow to record the real pH. For this reason, in my opinion, a table could be added with the assumed values without the interference of water.
- I suggest to add the pH values of the controls, as comparison with the treatments, in the paragraph 2.5 and 2.6 to better understand the differences.
- As for the discussion of the results the microbiota activity has been cited but without analyzing it, this should be indicated in the conclusions as a limitation of the study.
Author Response
Dear Reviewer: Thank you for your thoughtful suggestions and insights. The manuscript has been re-checked and the necessary changes have been made in accordance with the suggestions. The responses to all comments have been prepared and attached here with. Please see the attachment. Thanks again for your consideration. Sincerely Takeshi Murata
Response to Reviewer 1 Comments
Comment 1-1:
the authors analyzed only pH of female mice, and this should be indicated in the conclusions as a limitation.
A: Thank you for your helpful advice. We added a sentence about male animals in the conclusions section (Line 242).
Comment 1-2:
the addition of water, although made for all samples, did not allow to record the real pH. For this reason, in my opinion, a table could be added with the assumed values without the interference of water.
A: Thank you for your important suggestion. We added Tables 2 & 3 to show the pH values with a sign of inequality. Since purified water (approximately pH 7) was added to the contents, the real pH is higher for alkaline pH values and lower for acidic pH values. Table 1 was added to indicate the weights of contents used for the pH measurements, and it will now be clear how much sample was mixed with 50 μL water.
Comment 1-3:
I suggest to add the pH values of the controls, as comparison with the treatments, in the paragraph 2.5 and 2.6 to better understand the differences.
A: According to your suggestion, we added Tables 2 & 3 to show the pH values, to make it easier to compare the difference between the pH values in both Figures and Tables.
Comment 1-4:
As for the discussion of the results the microbiota activity has been cited but without analyzing it, this should be indicated in the conclusions as a limitation of the study.
A: Thank you for your kind recommendation. We added a sentence to show that the influence on microbiota was from cited information in the conclusions (Line 238).
Reviewer 2 Report
In their article entitled „Measurement of the intestinal pH in mice revealed alkalization induced
by antibiotics“ the authors have examined the pH of several sections of the gastrointestinal in mice differently treated. The main findings were that antibiotic treatment changes the pH.
An interesting study. However, the main flaws are the low numbers of mice used and the inappropriate statistical tests performed.
Here are my specific comments:
- Why was this title chosen, the authors have examined more than the influence of antibiotics on the pH of the murine gastrointestinal tract? Consider changing the title to more detailly reflect what you have studied.
- Introduction, line 34: Consider writing “Knowledge of the intestinal pH…”
- Results, line 50: Cecum or Caecum the authors should decide for one.
- Results, line 52: The authors write that the pH of the stomach was 3.41, in line 191 they state that the pH was not measured in the stomach.
- Results, line 54: This sentence is repetitive with the sentence before.
- Results, line 92: Again, the pH in the stomach is reported here and does not fit to the statement in line 191.
- Results, line 112: antibiotics do no “reduce” the intestinal microbiome.
- Figures: Which error bars are shown?
- Results, statements in lines 162ff: The authors state this as results but have not shown this with actual numbers and statistical tests. For instance, was the expansion measured?
- Methods: Experimental groups, four animals per group is a very low number
- Methods, line 190: I do not understand this statement. The authors have sacrificed four animals per group; why was it not possible to measure the pH in the stomach or jejunum?
- Methods: I think that the time of food intake and consequently the time of the experiments influences the pH in their gastrointestinal tract. Were the animals fasted before euthanasia? At what time of the day were the experiments done?
- Data Analysis: With four samples per group the authors cannot report mean and standard deviation. Were the data tested for normal distribution. Moreover, t-tests are the wrong tests for more than two groups.
- Conclusion, line 207: The microbiome was not measured, and again antibiotics do not reduce the microbiome, they may change it.
Author Response
Dear Reviewer: Thank you for your thoughtful suggestions and insights. The manuscript has been re-checked and the necessary changes have been made in accordance with the suggestions. The responses to all comments have been prepared and attached here with. Please see the attachment. Thanks again for your consideration. Sincerely Takeshi Murata
Response to Reviewer 2 Comments
Comment 2-1:
However, the main flaws are the low numbers of mice used and the inappropriate statistical tests performed.
A: Thank you for your comment. Please refer to the answer for Comments 2-11 and 2-14.
Comment 2-2:
Why was this title chosen, the authors have examined more than the influence of antibiotics on the pH of the murine gastrointestinal tract? Consider changing the title to more detailly reflect what you have studied.
A: According to your suggestion, we have changed the title to “Measurement of the Intestinal pH in Mice Under Various Conditions Reveals Alkalization Induced by Antibiotics”. We measured the pH in many conditions, and the influence of antibiotics is the main finding of this study. We think that the title is suitable for the journal “Antibiotics”.
Comment 2-3:
Introduction, line 34: Consider writing “Knowledge of the intestinal pH…”
A: Thank you. We corrected this sentence as you suggested.
Comment 2-4:
Results, line 50: Cecum or Caecum the authors should decide for one.
A: Thank you for your comment. We corrected this word as you suggested. We re-checked the words that are spelled differently between American and British English.
Comment 2-5:
Results, line 52: The authors write that the pH of the stomach was 3.41, in line 191 they state that the pH was not measured in the stomach.
A: Thank you for your important suggestion. We added some sentences in results 2.1 and Table 1 to explain this limitation (Line 51). Low amounts of contents (< 10 mg) were collected from the stomach and jejunum because mice were fasted overnight before euthanasia. The pH values of the stomach measured in this study serve as a reference and are not accurate compared to other pH values measured with enough sample.
Comment 2-6:
Results, line 54: This sentence is repetitive with the sentence before.
A: According to your comment, we deleted the repetitive sentence.
Comment 2-7:
Results, line 92: Again, the pH in the stomach is reported here and does not fit to the statement in line 191.
A: Please refer to the answer to Comment 2-5.
Comment 2-8:
Results, line 112: antibiotics do no “reduce” the intestinal microbiome.
A: Thank you for your helpful advice. We changed the expression as you suggested (Line 130).
Comment 2-9:
Figures: Which error bars are shown?
A: According to your comment about data analysis, four samples per group cannot be used to report the mean and standard deviation. Therefore, we plotted all pH values in the figures (Figures 2 & 4). For more details, please refer to the answer to Comment 2-14.
Comment 2-10:
Results, statements in lines 162ff: The authors state this as results but have not shown this with actual numbers and statistical tests. For instance, was the expansion measured?
A: Thank you for this important suggestion. We added Table 1 to indicate the approximate weights of contents collected from the GI tract. After antibiotic treatment or in GF mice, the contents collected from the cecum were approximately 1000 mg (more than three times higher compared to that of the control mice) (Line 188).
Comment 2-11:
Methods: Experimental groups, four animals per group is a very low number
A: We understand your concern; actually, four is a low number. We analyzed the data again with the correct method, as per your suggestion in Comment 2-14 (Line 230). As a result, the difference was still significant (Figures 2 & 4). We are trying to perform the additional measurements, but the Animal Research Ethics Committee of our university might not approve the additional experiments with respect to the reduction of experimental animal usage.
Comment 2-12:
Methods, line 190: I do not understand this statement. The authors have sacrificed four animals per group; why was it not possible to measure the pH in the stomach or jejunum?
A: Thank you for your question. Small amounts of contents were collected from the stomach and jejunum, because mice were fasted overnight before euthanasia (Line 51, Line 216). The ileum, cecum, and colon are relatively rich in commensal bacteria, and we collected more contents than that in the upper parts of the GI tract. It was thus difficult to measure the accurate pH value with less sample.
Comment 2-13:
Methods: I think that the time of food intake and consequently the time of the experiments influences the pH in their gastrointestinal tract. Were the animals fasted before euthanasia? At what time of the day were the experiments done?
A: Thank you for your kind recommendation. Mice were fasted overnight before euthanasia, and we performed the experiments in the morning (Line 216).
Comment 2-14:
Data Analysis: With four samples per group the authors cannot report mean and standard deviation. Were the data tested for normal distribution. Moreover, t-tests are the wrong tests for more than two groups.
A: You are quite right. In figures, we plotted all data in place of the mean and standard deviation. In main text and tables, we showed the data as the median (minimum value − maximum value). We attempted normal distribution, but some data were difficult to assess or did not follow a normal distribution. Therefore, we first treated data with a one-way analysis of variance (ANOVA) and compared them by Bonferroni’s correction with a non-parametric method (Line 230). As a result, differences between the mouse groups were still significant (Figures 2 & 4).
Comment 2-15:
Conclusion, line 207: The microbiome was not measured, and again antibiotics do not reduce the microbiome, they may change it.
A: Thank you for your helpful advice. We added a sentence to indicate that the influence on the microbiome is cited information in the conclusions and changed the expression as you suggested (Line 238).
Round 2
Reviewer 2 Report
The authors have performed some revisions.
However,still the authors use the wrong test for comparision. An ANOVA is used for normally distributed data and it is unclear which posthoc tests were used.
Additionally, I still think the number of animals is too low.